# Autophagy in Tumor Immunity and Viral-Based Immunotherapeutic Approaches in Cancer

**DOI:** 10.3390/cells10102672

**Published:** 2021-10-06

**Authors:** Ali Zahedi-Amiri, Kyle Malone, Shawn T. Beug, Tommy Alain, Behzad Yeganeh

**Affiliations:** 1Apoptosis Research Centre, Children’s Hospital of Eastern Ontario Research Institute, Ottawa, ON K1H 8L1, Canada; ali@mgcheo.med.uottawa.ca (A.Z.-A.); kmalo035@uottawa.ca (K.M.); shawn@arc.cheo.ca (S.T.B.); talain@uottawa.ca (T.A.); 2Department of Biochemistry Microbiology and Immunology, Faculty of Medicine, University of Ottawa, Ottawa, ON K1H 8L1, Canada; 3Centre for Infection, Immunity and Inflammation (CI3), University of Ottawa, Ottawa, ON K1H 8L1, Canada; 4Department of Cellular and Molecular Medicine, University of Ottawa, Ottawa, ON K1H 8M5, Canada

**Keywords:** oncolytic virus, autophagy, cancer treatment, cancer autophagy, immunotherapy, oncolytic immunotherapy, tumor microenvironment, tumor immunity

## Abstract

Autophagy is a fundamental catabolic process essential for the maintenance of cellular and tissue homeostasis, as well as directly contributing to the control of invading pathogens. Unsurprisingly, this process becomes critical in supporting cellular dysregulation that occurs in cancer, particularly the tumor microenvironments and their immune cell infiltration, ultimately playing a role in responses to cancer therapies. Therefore, understanding “cancer autophagy” could help turn this cellular waste-management service into a powerful ally for specific therapeutics. For instance, numerous regulatory mechanisms of the autophagic machinery can contribute to the anti-tumor properties of oncolytic viruses (OVs), which comprise a diverse class of replication-competent viruses with potential as cancer immunotherapeutics. In that context, autophagy can either: promote OV anti-tumor effects by enhancing infectivity and replication, mediating oncolysis, and inducing autophagic and immunogenic cell death; or reduce OV cytotoxicity by providing survival cues to tumor cells. These properties make the catabolic process of autophagy an attractive target for therapeutic combinations looking to enhance the efficacy of OVs. In this article, we review the complicated role of autophagy in cancer initiation and development, its effect on modulating OVs and immunity, and we discuss recent progress and opportunities/challenges in targeting autophagy to enhance oncolytic viral immunotherapy.

## 1. Introduction

Oncolytic viruses (OVs), either occurring naturally or through genetic engineering, are promising candidate therapies against cancer since they can selectively amplify in and kill tumor cells without affecting normal cells. Moreover, OVs can boost systemic anti-tumor immunity by lysing tumor cells; the consequent release of tumor-associated antigens (TAAs), damage-associated molecular patterns (DAMPs), and pathogen-associated molecular patterns (PAMPs) to the tumor microenvironment (TME) promote activation of antigen-presenting cells (APCs), in turn stimulating anti-tumor adaptive immune responses [1,2]. With their unique capacity to also encode for a vast array of therapeutic transgenes to enhance specific responses, OVs, therefore, represent promising candidate therapies against cancer.

Macro-autophagy (hereafter, autophagy) is a conserved and tightly regulated catabolic process in eukaryotic cells. Its main role is to supply energy during development, and under conditions of cellular stress, wherein selective or non-selective formation of double-membrane vesicles called autophagosomes engulf entire regions of the cytoplasm, superfluous or damaged organelles, protein aggregates, and invading pathogens (Figure 1). Whereas non-selective (bulk) autophagy is a cellular response to starvation and typically involves random uptake of cytoplasm into phagophores (the precursors to autophagosomes), selective autophagy is mainly responsible for removing certain components such as superfluous or damaged organelles, protein aggregates, and is driven by receptors with an affinity for both the cargo and LC3/ATG8 [3,4]. Autophagosomes are sequentially fused with lysosomes to form autolysosomes, whose contents are subsequently digested by lysosomal hydrolases and released for metabolic recycling [4]. This process is also used to eliminate foreign particles or infectious agents; however, like many other invading pathogens, many viruses, including OVs, have been known to interfere with the autophagic machinery either in cancer cells or cells of the TME to enhance their replication and spread [5,6,7,8,9,10]. This is important because as a component of the host defense system in cell-autonomous elimination of invading pathogens, autophagy can promote tumorigenesis or act as a tumor suppressor in different cancer models [11,12]. Further, autophagy regulates several immunological functions, such as inflammation activation, cytokine production, antigen, antigen processing, and T- and B-lymphocyte immune responses [13]. Thus, viral infection-induced autophagy may also contribute to anti-tumor immunity via enhancing processing and cross-presentation of tumor antigens by dendritic cells (DCs) [14,15], signifying that OV-mediated anti-tumor effects can be mediated by crosstalk between virus and host tumor and normal cells. Thus, a better understanding of the “host-virus-environment” in the context of autophagic regulation could help to efficiently manipulate their interconnections for more effective therapies.

We herein review autophagic mechanisms, their roles in cancer initiation and development, OV efficacy, and anti-tumor immune responses. Finally, the prospect and challenges of combinatorial OV therapy with autophagy modulators in cancer immunotherapy are explored.

## 2. Molecular Mechanism and Regulation of Autophagy

Although different signals are involved in initiating “selective” versus “non-selective” autophagy, both converge at the formation of the autophagosome upon membrane remodeling. Several factors come into play to initiate and sustain the system. Here we will describe the main known activation mechanisms that modulate the autophagic machinery; for visual representation, please refer to Figure 1. Activation of the Unc-51-like autophagy activating kinase 1 (ULK1) complex (ATG1 complex in yeast), which includes ULK1 and the subunits FIP200, ATG13, and ATG101, represents the initial hallmark of mammalian autophagy [16,17]. This occurs through autophosphorylation of ULK1 at the Thr180 residue, facilitated via kinase dimerization or oligomerization [18]. Upon sensing changes in cellular energy, the ULK1 complex acts as a scaffold to initiate the phagophore assembly site (PAS), a process triggered by binding of ATG13 to the early autophagy targeting (EAT) domain of ULK1 [19]. Localization of the ULK1 complex to the PAS can also be directed through interactions between members of the LC3 protein family and LIR/AIM short linear motifs of ULK1, which also have roles in later stages of autophagosome formation [20,21]. The binding of RAB1A, a small GTPase regulating vesicular trafficking, to the DENN domain (differentially expressed in normal and neoplastic cells) of C9orf72 is another essential factor contributing to ULK1 translocation to the PAS [22]. ULK1 phosphorylation of the phosphatidylinositol 3 kinase class III (PI3KC3) complex creates sub-complexes I (C1) and II (C2) and initiates autophagy. Common catalytic subunits of these complexes include ATG34, ATG15, and Beclin-1 (BECN1). PI3KC3-C1 is involved in membrane elongation and encompasses ATG14L that transports this complex to PAS. Interaction of PI3KC3-C2 with UVRAG then contributes to endosomal and autophagosome biogenesis [23].

Several protein-protein interactions and post-translational modifications regulate the kinase activity within PI3KC3-C1. For instance, NRBF2 promotes kinase activity and dimerization of this complex via interaction with ATG14L and BECN1 [24]. The binding of BCL2, an anti-apoptotic factor, to the BH3 domain of BECN1 represses the kinase activity of ATG34, resulting in inhibition of autophagy [25]. AMPBRA1, another PI3KC3-C1 regulator, can be phosphorylated by ULK1 to activate PI3KC3-C1 [26]. Following autophagy induction, the formation of the ATG12-ATG5 ubiquitination complex is catalyzed by ATG7 and ATG10 [27]. Once formed, this complex acts as a ubiquitin ligase to increase lipidation of LC3, a light chain of the microtubule-associated protein whose lipidation is essential for autophagosome formation and cargo recognition. The ATG12-ATG5 conjugate further interacts with ATG16L1 to establish the ATG12-ATG5-ATG16L1 complex [28], which is needed for activation of ATG3. This in turn facilitates LC3-phosphatidylethanolamine (PE) formation [29]. Although the exact role of the LC3-PE complex in autophagosome biogenesis is under investigation, in vitro findings suggest this complex is involved in membrane expansion and elongation of phagocytic vesicles for controlling the size and volume of autophagosomes, as well as in closure of these double-layer structures for completion of autophagy [21,29].

Numerous signal transduction pathways contribute to the regulation of autophagy. The main example is the mammalian target of rapamycin (mTOR) pathway, which functions as a serine/threonine protein kinase signaling regulating autophagy. Amino acid-triggered activation of mTOR complex 1 (mTORC1) on lysosomes can be mediated by several regulatory factors and complexes, such as RAG, SLC38A9, RAGULATOR-Rag, and KICSTOR [30]. AKT1, another protein kinase whose activation and phosphorylation could be stimulated by growth factors boosting PI3KC1, is also known to activate mTORC1 [31,32]. mTORC1 modulates the ULK1 complex, transcription factors, and phosphorylation of histone acetyltransferases. The second mTOR complex (mTORC2) can upregulate AKT1, which suppresses the activity of the FOXO3 transcription factor [33]. The transcription of autophagy-related genes such as *BNIP3* and *LC3* are in turn promoted by high levels of FOXO3, ultimately inducing autophagosome formation. AMP-activated protein kinase (AMPK) is another regulator of autophagy, of which activation of this kinase leads to increased autophagy via direct or indirect regulatory effects [34]. Direct regulation occurs through phosphorylation of components of autophagic complexes, including ULK1, PIK3C3, and mTORC1 [34,35]. Indirect effects involve alterations in autophagy-related gene expression.

Autophagic homeostasis can also be controlled by interactions between BECN1 and BCL-2. The downregulation of BECN1 reduces autophagy and promotes cell death under conditions of cell stress and nutrient deficiency [36]. In contrast, decreased BCL-2 levels substantially increase autophagy, also leading to cell death [37]. Moreover, upon disassociation from BCL2, BECN1 establishes various links with PI3KC3, resulting in the formation of inductive and activator complexes composed of BECN1, VPS15, and VPS34. The inductive complex includes ATG14L and can induce autophagy, whereas the activator complex activates both autophagy and endocytosis and contains UVRAG instead of ATG14L [36]. Among transcription factors, tumor protein 53 (TP53) plays a bidirectional and multilevel role in controlling autophagic activity. TP53-mediated increases in autophagy occur following upregulation of IGF-BP3, REDD1, and the AMPK pathway [38]. TP53-mediated autophagy initiation can also be triggered through targeting MAP1B, phosphorylated BECN1, and regulation of DAPK-1 [39]. Conversely, TP53 is also capable of inhibiting autophagy under certain circumstances via AMPK regulation. PTEN, DRAM, BNIP3, PUMA, BAX, BAD, and TIGAR represent other TP53 targets with autophagy-regulating capacity.

Several cellular responses mediate activation and modulation of the above-described factors to initiate the autophagic machinery. For instance, autophagy is regulated by endoplasmic reticulum (ER) stress, a cell-protective event that occurs in response to different intracellular and extracellular stimulating factors. ER stress promotes degradation of accumulated unfolded or misfolded proteins within the ER, triggering the unfolded protein response (UPR) and the ubiquitin-proteasome system (UPS) [40], with the goal of restoring ER homeostasis. On the other hand, high levels of ER stress-mediated autophagy can cause cell death [41,42]. Following the ER stress signal, autophagic vesicles partially engulf the damaged ER, leading to the reassembly of degraded fragments into new ER components [40]. Restoration of misfolded and unfolded proteins by UPR occurs via elevating protein expression, as well as activating ATF6, pancreatic endoplasmic reticulum kinase (PERK), and inositol-requiring enzyme 1α (IRE1 α) [43]. Restoration of ER homeostasis via UPR- and UPS-mediated autophagy requires persistent, strong stimulating signals.

Another potent cellular autophagic-inducing element is reactive oxygen species (ROS) generated during oxidative stress. Following exposure to oxidative stressors that overwhelm antioxidant capacity, an excessive redox state takes hold, with proteolysis, cellular damage, and DNA hydroxylation soon following. Lysosomal degradation pathways and autophagy can avert or delay cell death processes. Most ROS originate from the respiratory chain within the inner mitochondrial membrane. These act as autophagy inducers during oxidative stress, with high levels of ROS blocking the PI3K-Akt-mTOR pathway [44]. In vivo, PI3K-Akt-mTOR inhibition following ROS overproduction occurs through activation of AMPK [45]. ROS-triggered ATG4 deactivation subsequently increases LC3-II accumulation, promoting autophagosome formation. Elevated ROS, especially H_2_O_2_, has been found to block LC3-II following ATG4 oxidation, extending autophagic bodies [46]. ROS increases the degradation of ubiquitinated components by combining them with P62 and LC3-II for getting targeted and degraded by elongated autophagic bodies. ROS-mediated induction of autophagy can also occur via MAPK, in turn, governed by JNK, ERK, and p38 [47]. In murine mesenchymal stem cells, ROS have been found to induce autophagy via JNK signaling [48]. These regulate the expression of several autophagy-related genes by influencing AP-1, FoxO, and NF-κB. The p38 pathway also has roles in ROS-mediated autophagy, stimulating autophagosome-lysosomal fusion by increasing ROS levels and regulating ATG7 and E3 expression via ubiquitination [49].

Mounting evidence has revealed that several noncoding RNAs regulate autophagy. For example, microRNAs (miRNAs), which are short noncoding RNAs controlling gene expression, were shown to regulate different steps of autophagy, from induction to fusion. During the initiation stage, different miRNAs act on ULK1/2, mTOR, and FIP200, thereby controlling the induction of autophagy [50]. Entirely different varieties of miRNAs were found to control the nucleation step of the autophagy pathway via altering the expression of BECN1, AMBRA1, UVRAG, and ATG14 [51]. miRNA-related regulation of the autophagosome elongation stage involves ATG12, ATG5, and ATG7, while the autolysosome fusion step can be affected by miRNAs regulating RAB7, LAMP2, RAB27A [52,53]. In addition to miRNAs, various steps of autophagy pathways can be altered by long noncoding RNAs (lncRNAs), as well as non-protein-coding RNA transcripts [54].

Other mechanisms that regulate autophagy pathways, among others, can include DNA methylation, histone modifications, and the NF-κB pathway, although research into these regulatory pathways is still in its early stages [55,56]. In the next sections, we address cancer immunity, immunotherapy, and alterations in autophagy and its regulatory mechanisms, particularly mediated by different types of OVs.

## 3. Autophagy in Tumor Initiation and Development

Depending on the cancer stage, type, and components of the TME, autophagy acts in a double-edged manner during tumor initiation and growth. Table 1 provides a summary of the dual role of autophagy in tumor promotion and inhibition in different cancers. As a degradation system for recycling damaged organelles and dysfunctional proteins, autophagy prevents chronic tissue damage and the aggregation of oncogenic p62 protein under stress conditions, ultimately inhibiting not only the formation of tumors but also their proliferation, invasion, and metastasis during the early stages of tumorigenesis [57]. Conversely, in late-stage tumors, autophagy protects cancer cells against factors such as DNA damaging agents, nutrient and growth factor starvation, and hypoxia, thereby enhancing tumor growth and development [12]. Therefore, defining ways to activate or reverse autophagic mechanisms could provide avenues to control cancer progression and is currently an active field of research.

First and foremost, mutations in autophagy regulator genes have long been known to be associated with various malignancies. For instance, tumorigenesis might occur following abnormal expression of proteins controlling autophagy such as BECN1, considered a tumor suppressor monoallelically found in several types of tumors while downregulated in others, such as hepatocellular carcinoma [58]. In vivo studies have reported that *BECN1^+/−^* mouse models more readily develop tumors [59,60]. Other autophagy-related genes with identified roles in tumorigenesis include *ATG2B*, *ATG5*, *ATG9B,* and *ATG12*; frameshift mutations in each have been detected in patients with colorectal and gastric cancers [61]. *ATG5* point mutations were detected in biopsies of patients with hepatocellular carcinoma, gastric and colorectal cancers, suggesting the potential involvement of ATG dysregulations in abnormal autophagy and tumor development [62]. Accumulation of p62 has also been demonstrated in various types of cancer, including breast, prostate, lung adenocarcinoma, and gastrointestinal cancer [63]. Elevated p62 is considered a marker of tumor progression. During the normal process of autophagy, p62 is naturally degraded after interaction with LC3, a process that inhibits tumorigenesis [64]. In tumors progressing to late stages, autophagy can maintain cancer cell survival and growth, and one such mechanism through which this is achieved is overexpression of *BECN1* rather than reduced expression, then contributing to tumor growth and survival [65]. Degradation of dysfunctional mitochondria represents another way in which autophagy promotes cancer cell resistance to cell stress and cytotoxic stimuli [66]. Similarly, autophagy-triggered suppression of necrosis and inflammatory cell infiltration inhibit the initial stages of epithelial-to-mesenchymal transition and can participate in promoting metastasis [67]. In contrast, later stages of metastasis are enhanced by autophagic processes, facilitating the spread of cancer cells into the circulation, colonization of targeted organs, and activating the latency of metastatic cells [68]. After colonization of destination organs, autophagic flux is activated to protect against hypoxia, nutrient deficiencies, and segregation from extracellular matrix [69,70]. Several studies have reported upregulation of LC3B in metastases of different cancers, including hepatocellular carcinoma, melanoma, glioblastoma, and breast cancer, illustrating a key role for autophagy in promoting metastasis [71,72,73,74].

## 4. Autophagy in Tumor Immunity

Despite oncogenic mutations and their consequent effects on normal cellular processes generating unique, novel antigens (neoantigens) that differentiate them from normal cells, effective immune responses against cancer are rare. A tenuous equilibrium develops between the tumor and immune system wherein susceptible cancer cells expressing neoantigens are cleared, survived by those lacking neoantigen expression or possessing other mechanisms of immune evasion, such as increased immune checkpoints or loss of MHC expression. This process is referred to as immune editing [85]; the immune system becomes a vital component of cancer progression [86,87]. Disrupting this equilibrium is the goal of targeted cancer immunotherapies, and with its critical role in mediating homeostasis, it is clear that autophagic mechanisms directly participate in regulating responses to anti-tumor immune responses. Active immunotherapies look to directly stimulate long-term anti-cancer immunity (for example, cancer vaccines), whereas passive immunotherapies produce transient increases in the anti-cancer immune response, requiring multiple doses (for example, monoclonal antibody treatments) [88]. Below we will discuss these approaches taking into consideration the role of autophagy within cancer cells, the tumor microenvironment, and its immune profile.

There are three broad phases of immunoediting: (1) elimination, (2) equilibrium, and (3) escape [89]. First, cells with defective DNA repair mechanisms either die or are killed by the innate and adaptive immune system through normal immunosurveillance, of which autophagy mechanisms have been known to contribute (reviewed in the work of [90]). Tumor cells evading immune destruction establish equilibrium with immune cells, which edit tumor growth and immunogenicity. Finally, these edited tumors escape via upregulation of multiple immunosuppressive mechanisms and continue to proliferate unchecked [88,89]. Cancer stem cells (CSCs) represent a subpopulation within the tumor bulk that are thought to be responsible for tumor initiation, immune evasion, resistance to therapy, and relapse and are capable of adopting “dormant” and proliferative states [91,92]; this concept of CSC dormancy is reviewed in [93]. Recent findings have elucidated the role of autophagy in maintaining the CSC population [94]; among the survival mechanisms employed by “dormant” cells, autophagy and senescence have emerged as key resistance mechanisms to apoptotic triggers [95]. For instance, autophagy can be initiated in prostate cancer cells exposed to IL-6 and IL-2, while anti-inflammatory IL-10 blunted autophagic responses [96,97]. IL-24 signaling likewise induces autophagy in melanoma cells via PI3K-mediated BECN1 induction [98]. Intratumoral cell-cell contacts can induce autophagy in infiltrating lymphocytes [96]; co-therapies with chloroquine, an inhibitor of autophagy, enhance immunotherapies against melanoma and renal cell carcinoma [99].

As the initial, non-specific responders to viral and bacterial pathogens and cancer and a vital component of an effective adaptive immune response, the innate immune system is an important target for enhancing adaptive, T cell-based cancer immunity; the role of B cells is less well defined [87]. In addition, given the role of OVs in tumor cell death, the initial response and potential resistance mechanisms mediated by the innate immune system are of particular concern for oncolytic virotherapy. The first three steps of the cancer immunity cycle are performed primarily by DCs. Autophagy plays a key role in DC antigen presentation on MHCs, seeming to enhance MHC-II presentation [14] and, through inhibition of MHC-I externalization, reduce MHC-I presentation and subsequent CD8^+^ T-cell stimulation [100]. The role of autophagy in all processes of the innate immune system is reviewed in [101,102]. While experimental work has shown that plasmacytoid DCs (pDCs), activated by Toll-like receptors (TLRs), are able to secrete type 1 IFNs, prime T cells and coordinate anti-tumor immune responses [103,104,105], their infiltration into the tumor is accompanied by the adoption of an immunosuppressive phenotype; high pDC levels are associated with a worse prognosis [106]. Autophagy has been found to play a role in all aspects of DC function (reviewed in [107]). Briefly, autophagy seems to play an inhibitory role in DC maturation [108] and migration [109] while acting upstream to enhance TLR signaling [110]. Autophagy was found to be required for TLR9-induced IFN-α production by pDCs [111], and similar roles for autophagy have been found for DC production of TNF-α, IL-12, and IL-6 and subsequent T-cell responses [112]; DC ATG5 activity is required for CD4^+^ T-cell production of IFN-γ and IL-2 [113], and DC autophagy is required for T-cell activation [114], including regulatory T cells (Tregs) [115]. Interestingly, conflicting findings in regards to DC autophagy and Tregs have been found; Clement et al. noted in atherosclerosis models that disrupted DC autophagy results in Treg expansion [116]. Contrasting roles have also been found for cytokine production; IL-10 production requires functional DC autophagy [117], while Tregs inhibit DC autophagy, contributing to the resolution of inflammation [118]. The other major DC subtype, conventional DCs (cDCs), display more anti-tumor activity. The CD8α^+^ (mouse) and CD141^+^ (human) type 1 cDC (cDC1) subtype efficiently presents major histocompatibility complex (MHC) class 1 antigens to CD8^+^ T cells, initializing potent anti-tumor immune responses in “hot” tumors [119,120]. Contrary to pDCs, increased cDC1 infiltration is associated with an improved prognosis and more effective anti-tumor immune responses [121,122].

Tumor-associated macrophages (TAMs) represent another early innate immune cell with key roles in tumor development, recruited by chemokines released by tumor cells and associated stroma in a colony-stimulating factor 1 (CSF-1)-CSF-1R-dependent manner [123,124]. A crucial role for autophagy in the differentiation of monocytes into macrophages has been described [125]. TAMs play vital roles in every aspect of tumor progression [87]. Initially, their cytokine secretion profile induces low-level inflammation mediated by IL-1β and IL-6. Later, they play crucial roles in tumor cell migration and angiogenesis via growth factor and matrix metalloproteinase secretion [126,127]. Autophagy has been found to modulate phagocytosis [128,129] and reduce inflammasome activation and IL-1β expression [130,131]. TAMs can assume a spectrum of differentiation phenotypes, from the classically activated pro-inflammatory M1 to the alternatively activated, anti-inflammatory M2, each with subtypes characterized by unique transcriptional and secretory features [132]. The majority of TAMs display an M2-esque, anti-inflammatory phenotype, induced through IL-4, IL-13, and IL-10 signaling and, through the secretion of anti-inflammatory factors such as TGF-β and CCL22 (which recruits Tregs) and upregulation of immune checkpoints, inhibit anti-tumor T-cell responses [133,134] and promote many aspects of tumor biology, including migration and angiogenesis [135,136]. As a major mechanism of tumor resistance to immune attack, conversion of M2 TAMs into a pro-inflammatory M1-like phenotype (or reduction in the overall TAM population) has become a feature of many combination immunotherapies, aiming to enhance T-cell attack. This may act to reduce Treg immunosuppressive functions, and targeting this population also represents a feature of many immunotherapies. Tregs (CD4^+^CD25^+^FoxP3^+^) are the most potent anti-inflammatory cell in the body and act to prevent autoimmunity, which, in the context of cancer, significantly limits anti-tumor immunity. M1 macrophages are characterized by lower levels of autophagy [137], and autophagy is generally considered anti-inflammatory [101]. The mechanisms through which these cells suppress inflammation include secretion of anti-inflammatory cytokines (IL-10, IL-35, TGF-β), induction of naïve and effector T-cell apoptosis, inhibition of DC function and maturation, immune checkpoint expression, inhibition of T-cell co-stimulation, and T-cell receptor signaling through connexin channels, and metabolic disruption, and inhibition of naïve T-cell proliferation through high CD25 expression that sequesters IL-2 [138,139,140].

Another key member of the early immune response to tumors is natural killer (NK) cells. NK cells possess natural cytotoxicity receptors that respond to ligands associated with DNA damage and cellular stress, such as NKp30, NKp44, and NKp46, NKG2D, -2C, and DNAM-1 [141]. Discrimination of normal vs. tumor cells is achieved through sensors of MHC class I (MHC1), with NK cells preferentially killing MHC1^-^ cells via the release of perforin and granzymes. Cells lacking MHC1 are sensed primarily through killer-cell immunoglobulin-like receptors (KIRs) and NKG2D [141,142]. Autophagy is required for NK cell development, differentiation, survival, cytolytic functions, and memory responses [143,144]. Communication between NK cells and DCs increases the frequency of cDC1s while further promoting tumor cell death, antigen capture, presentation, and T-cell activation; high NK cell infiltration is associated with improved prognosis and immunotherapy responses [145,146]. Autophagy blockade in melanoma models resulted in increased expression of CCL5, increasing intratumoral NK cell infiltration [147]. As with both TAMs and DCs, however, factors within the TME restrict NK cell activity and can lead to NK cell exhaustion, such as that seen in T cells (reviewed in [148,149]). While certain tumors limit NK cell entry through restrictions to effective homing and entry, within the TME immunosuppressive stroma, Tregs and TAMs (including myeloid-derived suppressor cells, MDSCs) limit their cytotoxic capacity. High levels of immune checkpoints [150,151] and upregulation of enzymes such as indoleamine 2,3-Dioxygenase (IDO1) and inducible nitric oxide synthase (iNOS) within a tumor restrict the available pool of amino acids required for NK cell (and T-cell) effector functions [141,152,153]. Hypoxia, a key feature of many tumors, is sensed through hypoxia-inducible factor-1 and -2 (HIF-1 and -2). Increased HIF signaling limits DC antigen presentation and effector T-cell migration, proliferation, and cytotoxic capacity while in turn enhancing TAM retention, Treg development, and the immunosuppressive actions of both [154,155]. Natural killer (NK) cell cytotoxic effects are reduced, while MDSC numbers and actions are enhanced. Hypoxia is able to trigger tumor cell autophagy through HIF1-BNIP3-BECN1 signaling, clearing damaged mitochondria and reducing ROS production [156], contributing to enhanced survival. Tumor cells secrete increased levels of anti-inflammatory cytokines and exhibit reduced major histocompatibility complex and increased immune checkpoint expression [157], contributing to the reduced NK cell effector function and efficacy of checkpoint blockade therapies. Therapies looking to enhance the anti-tumor effect of NK cells are currently receiving significant attention, looking to remove inhibitory signals and enhance NK cell activation (reviewed in [141]) as a potential alternative, or in combination with T cell-based therapies, depending on the tumor. Given the seeming ubiquitous role of autophagy in NK cell processes, the potential of broad autophagy blockade in vivo would produce similar results is unlikely, suggesting the necessity for tumor cell-targeted therapies.

While the diversity of immunosuppressive mechanisms within a tumor may limit the efficacy of monotherapies, targeted combinations against multiple inflammatory and cell death pathways, including the autophagic machinery, as well as against both arms of the immune system, are likely to represent the most effective way forward in cancer immunotherapies. Among these pathways, as mentioned previously, autophagy can represent a double-edged sword, regulating a plethora of inflammatory functions and immune phenotypes while also promoting tumor cell survival [158,159,160]. The exact role of autophagy in each context, especially in a site as metabolically complex and genetically unstable as a TME, will be crucial to elucidate prior to its effective use in targeted immunotherapies.

## 5. Targeting Autophagy for Immunotherapy

The cancer immunity cycle, outlined by Chen and Mellman [161], describes seven key steps in the anti-tumor immune response, which in turn can be therapeutically targeted (Figure 2). Briefly, these steps are: (1) neoantigen capture and processing by antigen-presenting cells (APCs), most notably dendritic cells (DCs); (2) trafficking of DCs to lymph nodes for antigen presentation to lymphocytes; (3) lymphocyte priming and activation upon antigen presentation; (4) trafficking of activated T cells through the blood to the tumor location; and (5) out of circulation into the tumor itself, wherein; (6) T cells recognize cancer cells expressing neoantigens and; (7) effect tumor cell killing, with consequent antigen capture and processing by APCs, starting the cycle anew [161]. Each of these steps can be modulated by the autophagic response.

Given the importance of the immune system and the autophagic response in cancer progression, repurposing the established inflammatory milieu to permit enhanced T-cell attack remains the major goal of immunotherapy. Two broad categories of tumors have been defined based on relative immune infiltration. Type 1, T cell-inflamed (“hot”) tumors, with high tumor mutation burden, spontaneous T-cell activation, and infiltration, along with general expression of chemokines, activated APC markers, and type 1 IFN responses. These escape immune elimination via increases in immune checkpoint expression, reductions in neoantigen expression, resistance to cytotoxic molecules, and a metabolic microenvironment that hinders effective immune responses [89,126]. The second type 2 category is the non-T cell-inflamed (“cold”) tumors, lacking any such markers, with low mutation burden and characterized by little to no T-cell infiltration upon immunohistochemical analysis [89,126]. Other types of tumor microenvironments (TME) have also been described, defined by intermediate levels of inflammation and tumor mutation burden [89]. These subtypes point to unique immune defects between tumor types, wherein “hot” tumors fail at the effector phase and “cold” tumors display immune exclusion [126], generally corresponding to late and early stages of the cancer immunity cycle, respectively. Autophagy has key roles in both stages (reviewed in [162]). Relating to the initial steps, the release of DAMPs, recognized by innate immune cells that consequently initiate immune responses, occurs following cancer cell death. The pro-survival effects of autophagy would reduce, while autophagic cell death would increase, DAMP release [163]; thus, autophagic regulation of cancer cell survival is key to mediating early stages. DAMPs, in turn, regulate autophagic signaling [162]. Inhibition of autophagy via ATG5 deletion has been found to increase intratumoral anti-inflammatory Tregs [164], contributing to the suppression of cytotoxic T-cell responses and inhibiting effector functions, corresponding to control of late stages of the cycle. Interestingly, inhibition of autophagy improves the efficacy of anti-PD-1 and anti-CTLA-4 checkpoint therapy in normally resistant pancreatic ductal adenocarcinoma models [165], as well as anti-PD-1 treatments of murine prostate cancer [166]. Conversely, activation of autophagy can lead to degradation of PD-L1 and CTLA-4, allowing for enhanced immune-mediated tumor destruction [167]. Therefore, determining the most effective means of regulating autophagy is likely to provide significant benefit to immunotherapies targeting any stage of the cancer immunity cycle.

Aside from direct effects on tumor cells, autophagic modulators also affect systemic immunity, likely a result of the multifunctionality of many pathways crucial to autophagy. As with the double-edged role in cancer treatment, autophagic modulators have multifaceted effects on the immune system; in general, however, they are immunosuppressive. Rapamycin, an mTOR inhibitor, has been found to enhance γδ T-cell cytotoxicity and anti-cancer effects [168]. In models of influenza infection, rapamycin has been found to reduce B-cell class switching, proliferation, formation of germinal centers, and antibody specificities. Further, populations of memory CD8^+^ and CD4^+^ T cells, including Tregs, are increased by rapamycin treatment [169]. Chronic, long-term mTOR inhibition using rapamycin has been found to extend mouse lifespan; total T-cell populations were reduced in the spleen, and a preference for naïve phenotypes was observed [170]. In models of bone marrow failure, similar reductions in effector and memory T-cell populations and a preference for naïve phenotypes upon rapamycin treatment have been observed, with consequent improvements in mouse outcomes [171]. Following chronic rapamycin treatment, T-cell trafficking markers (and trafficking capacity) were increased, while checkpoints and exhaustion were decreased. Conversely, Treg populations were increased, as were myeloid cell populations, with alterations to DC-T-cell interactions [170]. mTORC1 inhibition increased myeloid cell stimulation of TH1 or TH17 inflammatory responses, including increased IFN-γ and IL-17 production [172]. Rapamycin expands natural and induced Treg populations and enhances DC migration while inhibiting their maturation while also reducing differentiation of MDSCs [173].

Chloroquine, another modulator of autophagy, is often used to treat autoimmune diseases given well-characterized immunosuppressive activity (reviewed in [174]). Briefly, hydroxychloroquine and chloroquine dampen TLR signaling and interfere with antigen processing, MHCII expression, and presentation with a consequent reduction in T-cell activation, cytokine production, and effector functions [174,175]. In mouse models of malaria, chloroquine significantly suppresses macrophage, B-cell, and helper T-cell activation while dampening DC maturation [176]. Inhibition of many aspects of innate immunity leads to reduced type I IFN and other pro-inflammatory cytokine responses, including TNF-α, IL-6, and IL-1β, which result in dampened adaptive immune responses [177]. Similar immunosuppressive effects have been noted for vorinostat, a histone deacetylase inhibitor. Reduced NK cell cytotoxic effects [178], enhanced Treg immunosuppression [179,180], and increased DC expression of IDO1 [181]. Interestingly, levels of anti-inflammatory IL-10 are reduced following treatment with vorinostat [182].

In summary, while autophagy blockade may induce immunogenic cell death, and its anti-cancer effects are at least partially dependent on a functional immune system, much of the immunomodulatory effects of current pharmacological agents are immunosuppressive, illustrating a need for improved understanding of autophagic processes and more targeted therapies.

## 6. Autophagy and OVs

OVs are defined as modified or natural viruses that preferentially replicate within cancer cells to induce cell lysis. OVs trigger immunogenic cancer cell death with the consequent release of TAAs and DAMPs. Ultimately, these stimulate APCs and T cell-based adaptive immune responses toward tumor antigens [1,2]. Given this, OVs represent a promising avenue of cancer immunotherapeutic, with herpes simplex virus (HSV), reovirus (REO), measles virus (MV), retroviruses, myxoma virus (MYXV), coxsackievirus, parvoviruses, adenovirus (Ad), vesicular stomatitis virus (VSV), vaccinia virus, polioviruses, Newcastle disease virus (NDV), and Seneca Valley virus (SVV) translated into clinical trials, alone or in combination therapies [183]. Autophagy plays a dual role in oncolytic virotherapy. Although the effects of viruses on normal autophagy appear to be both virus- and cell-specific, replication of OVs can be promoted by autophagy, enhancing tumor cell death. Paradoxically, autophagy can also metabolically nourish cancer cells, protecting them from OV-mediated oncolysis and cytotoxicity. Irrespective of autophagy modulation toward oncolysis, OVs themselves can also induce different types of cell death in cancer cells through various mechanisms. For example, Ad infection was reported to mediate necroptosis via upregulating RIP3 and promoting the phosphorylation of mixed-lineage kinase domain-like (MLKL) in lung cancer cells [184]. However, in malignant pleural mesothelioma cells, Ad triggers immunogenic cell death by overexpressing HMGB1 and CRT [185]. These cell-specific differences in response to cell death have also been discovered to occur following infection with other OVs, such as vaccinia, HSV, and NDV. Several studies have demonstrated that certain OVs are capable of altering autophagic signaling to enhance their replicative and lytic capacity while simultaneously suppressing antiviral innate immune responses. Below we will describe reports of specific OVs and their interaction with the autophagy response in modulating therapeutic efficacy.

Ad uses autophagy to improve its spread among cancer cells (Figure 3). Rapamycin, an autophagy promoter, and immunosuppressant, has been shown to increase replication of chimeric Ad5/F35, while autophagy inhibition using 3-methyladenine (3MA) restricts Ad replication, as well as production of viral structural proteins [7,186]. Conversely, treatment of glioma cells with the oncolytic Ad d1922–947 was characterized by Akt/mTOR activation, Erk1/2 inhibition, and reductions in autophagic flux. Furthermore, chloroquine-mediated inhibition of autophagy in these cells has been found to reduce Ad genome copies [187]. Autophagy induction by Ad may enhance cell lysis without concurrent increases in replication. An example of this phenomenon has been observed in glioblastoma cells after infection with the OBP-405 Ad virus, where neither autophagy activation by temozolomide and rapamycin nor targeted siRNA and 3-MA-mediated inhibition altered viral replication. The level of oncolysis was counter-intuitively increased through autophagic activity [9]. OBP-301, another Ad, is capable of causing autophagic cell death through miR-7-induced EGFR inhibition [188]. Similarly, in leukemia cells, infection with BECN1-equipped chimeric oncolytic Ad was noted to cause robust autophagic cell death in vitro and in vivo [189]. In addition to autophagic cell death, it has been shown that combining oncolytic Ad with temozolomide not only decreases tumor growth but evokes immunogenic cell death via upregulation of high mobility group box protein 1 (HMGB1), CRT, and ATP, which collectively stimulate various anti-tumor immune responses [190]. These include phagocytosis by DCs, elevated cytotoxic T lymphocytes, and increased expression of pro-inflammatory cytokines [191]. As discussed earlier, due to the paradoxical mechanism of action in respect to various tumors, autophagy could restrict the oncolytic capacity via protecting and nourishing cancer cells. For instance, in ovarian cancer cells, Ad-triggered autophagy was reported to inhibit the oncolytic activity of this OV, while autophagy suppression by 3-MA and chloroquine appears to increase the cytotoxicity mediated by Ad [192]. Similar results were obtained in U373MG and U87MG glioma cell lines following inhibition of autophagy and infection with Ad [187].

Attenuated MV has been found to induce autophagy to enhance viral replication (Figure 4). In HeLa cells, MV binds CD46, which contains a cytoplasmic domain linked to the BECN1 complex through the scaffold protein GOPC. During autophagy induction, the C protein of MV plays a fundamental role in increasing viral replication and production of viral progeny particles by delaying cell death [5]. One mechanism through which MV-C protein triggers autophagy is via interacting with immunity-related GTPase family M (IRGM), a protein that influences autophagy master regulators such as LC3, ULK1, ATG5, and ATG10, in addition to phosphorylating BECN1 [193,194,195]. Oncolytic MV can also use mitophagy to downregulate mitochondrial antiviral signaling (MAVS), consequently suppressing innate immune responses activated by RIG-I-like receptors and elevating MV replication in tumors [8].

Replication of oncolytic NDV can similarly be boosted by autophagy (Figure 5). In NDV-treated glioma cells, chloroquine treatment or siRNA-mediated downregulation of BECN1 and ATG5 significantly reduce viral replication, whereas rapamycin-induced activation of autophagy negatively affects NDV dissemination [196]. Unlike Ad, NDV leads to glioma cell autophagy activation, subsequently mediating immunogenic cell death through increased production of CRT, PMEL17, and HMGB1 [183]. Moreover, in mouse glioma models, intratumoral NDV administration enhanced infiltration of IFNγ^+^ T cells into the TME, suggesting increased immunogenic cell death [197]. In lung cancer cells, autophagy induction by NDV has been demonstrated to increase levels of HSP70, which interacts with its receptor on APCs to stimulate CD8^+^ T cell responses and activate NK cells [10] (Figure 5). NDV-infected melanoma cells were also found to express higher levels of immunogenic cell death regulators such as HSP70 and HMGB1. Inhibiting autophagy in these cells can result in reduced expression of these markers, highlighting the role of NDV-mediated autophagy in enhancing immunogenic cell death of tumor cells [198].

VSV also takes advantage of autophagy for intratumoral expansion (Figure 6). NRF2-induced autophagy, mediated through overexpression of HO-1, has been found to increase the replication of oncolytic VSV while suppressing VSV-triggered immune responses, achieved by dysregulating the pathway controlling RIG-I-MAVS interactions [199]. A combination of VSV with different autophagy regulators was also revealed to boost VSV replication and lytic capacity. For instance, in pancreatic cancer cells, co-treatment of oncolytic VSV and vorinostat upregulated the expression of some ATGs via hyperacetylation of NF-κB. In contrast, blocking autophagy using 3-MA increases the activity of IFN stimulated genes, resulting in decreased oncolytic capacity and restricted VSV replication [200]. Further, *BECN1* knockdown appears to results in HeLa cell sensitivity to VSV-induced oncolysis, illustrating the role of autophagy in altering cancer cell resistance to oncolytic virotherapy [201].

Among other OVs mediating or exploiting autophagy, HSV-1 enhances cytoplasmic aggregation of LC3 and autophagosome formation, causing autophagic cell death in squamous cell carcinoma [6]. HSV-1 replication does not seem to be influenced by autophagy inhibition, but its lytic capacity and cytotoxicity are reduced when autophagy is inhibited [183]. Conversely, disrupted or abnormal autophagy has been found to increase both the replication and anti-tumor activity of oncolytic SVV in orthotopic xenograft models [202]. Autophagic cell death can be seen in leukemia and myeloma after infection with engineered oncolytic vaccinia expressing BECN1, which elevates viral replication [203]. Oncolytic reovirus and some of its derived formulations have also been shown to use autophagy. Pelareorep is one example that was recently demonstrated to increase the expression of autophagy-related proteins, such as LC3 and ATG5, in colorectal cancer cells, a phenomenon that boosts not only the propagation of reovirus but also cell death [204]. Avian reovirus p17 non-structural protein upregulates AMPK and PTEN while suppressing mTORC1, eventually activating autophagy [205]. However, the exact mechanism through which mammalian reovirus induces autophagy remains largely undiscovered. Taken together, the results of different studies suggest virus-specific and cancer-specific alterations in autophagy pathway and cell death mechanism after infection of various cancer cells or tumors with OVs.

## 7. Targeting Autophagy in Oncolytic Immunotherapy

Aside from the aforementioned pharmacological modulations, autophagy is regulated by numerous upstream molecules that represent potential targets for altering autophagic signaling in order to enhance oncolytic immunotherapy. PERK and IRE1α, which are involved in ER stress responses, are promising targets [206]. Li et al. have demonstrated that STF083010, a nuclease inhibitor of IRE1α, promotes accumulation of oncolytic M1 alphavirus in glioma cells in vitro and in vivo via suppression of autophagic degradation [207]. Interestingly, the M1 virus itself is able to induce autophagy in glioma cells, a phenomenon that subverts effective oncolysis. Reduced IRE1α had no influence on M1-triggered STAT1 phosphorylation, with no effect on type I IFN signaling and innate antiviral response following infection [207,208]. IRE1α inhibition also had no effect on M1 viral replication. In addition to glioma, siRNA-mediated knockdown of IRE1Α has been found to increase cell death of leukemia cells through PERK-dependent autophagy [209]. These results suggest that targeting IRE1A may sensitize cancer cells to OVs through either autophagy-dependent or -independent mechanisms, depending on tumor stage and type. The detailed effects of IRE1A manipulation on autophagy require further investigation.

UPR is another candidate target in modulating autophagy to enhance virotherapy. UPR activation occurs after severe ER stress (e.g., following a persistent infection) that cannot be neutralized by tumor cells, inducing cell death; given this, UPR has the capacity to kill OV-infected cancer cells. This UPR-triggered cell death occurs upon failure of UPR to restore cellular homeostasis; inefficient UPR function can be compensated for by activating autophagic degradation to remove the ER and other damaged organelles [210]. The reciprocal communication between autophagy and UPR might play a fundamental role in directing cell fate decisions in tumors after exposure to OVs. For instance, UPR inhibition via IRE1α silencing can increase maraba virus-induced cytotoxicity in glioblastoma cells [211]. In contrast, oncolysis of melanoma and lung adenocarcinoma cells following oncolytic adenovirus infection was enhanced following downregulation of Golgi-specific brefeldin A-resistance guanine nucleotide exchange factor 1 (GBF-1), which activates UPR [212]. Although not investigated, UPR suppression is thought to inhibit autophagy, while its activation is inducing autophagy. Some OVs can naturally induce UPR and consequently autophagy; VSV was discovered to increase UPR expression markers such as ATF4, BIP, and eIF2α, in fibrosarcoma cells [213], as has M1 virus similar to maraba virus, siRNA-mediated suppression of UPR regulators promotes M1 virus-induced oncolysis in U87 glioma cells, a condition accompanied by inhibition of autophagy [207]. Different OVs and tumor cells need to be tested to fully elucidate mechanisms of oncolysis and autophagy upon UPR-targeted therapies.

In summary, whether targeting the ER stress or UPR pathways produce the ideal effect on autophagy for enhancing OV therapies will depend on both the type of tumor and OV used. Preferably, such alterations would not only lead to increased replication and viral protein synthesis but also enhanced immunogenic cell death and oncolysis. Considering OV-specific differences in using ER and UPR, more screenings should be implemented to characterize the vulnerability of various OV-infected cancer cells in response to dual aspects of autophagy regulation by these pathways. Additionally, several other master regulators and interacting molecules governing ER stress and UPR might have the capacity to modulate autophagy through other mechanisms, which could reveal new intracellular targets for improving oncolysis and anti-tumor immune responses.

## 8. Concluding Remarks and Future Perspectives

The efficacy of targeting autophagy in oncolytic virotherapy relies on several factors, including OV variants, cancer cell types, and effective induction of anti-tumor immune responses. In cell-specific and OV-specific manners, both activation and inhibition of autophagy appear to boost oncolysis via the mediation of immunogenic cell death. Such dual functions make it difficult to target autophagy unilaterally for enhancing OV propagation in a variety of tumors. Consistently, autophagy also appears to have pro- and anti-inflammatory effects within the TME, depending on context and immune cell type. Accordingly, a precise understanding of the molecular interaction between autophagy, the immune system, and OV biology could open new avenues for optimizing oncolytic viral and other immunotherapy. Nevertheless, many obstacles remain, including off-target effects of pharmacological agents within the TME and the instability of autophagic processes at various stages of tumor development (e.g., acting as both tumor suppressor and tumor survival factor), altering the optimal timing for targeting autophagy and affecting the efficacy of oncolytic virotherapy. Various recombinant systems and inducible alleles allow for in vitro and in vivo modeling of OV-infected cancer cell responses to agents targeting autophagy during different stages of tumor development and progression. Recent advances in metabolomics allow for accurate determination of the role of autophagy in tumor initiation, progression, and relationship with the immune system. Overall, research will determine the safety and efficacy of using autophagy modulators in future oncolytic immunotherapy.

## Figures and Tables

**Figure 1 cells-10-02672-f001:**
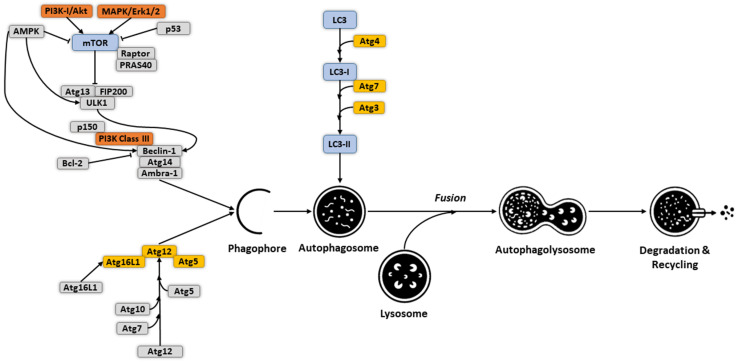
Molecular mechanism of autophagy activation. The kinase mTOR is known as the master regulator of autophagy initiation. In nutrient-rich conditions, the mTOR kinase associates with the ULK1 complex to inhibit the initiation of autophagy. However, under growth factor deprivation, nutrient starvation, or other stimuli, activation of energy sensor AMPK or TP53 (p53) leads to activation of the ULK1 complex and inhibition of mTOR activity through phosphorylation. Phosphorylated ULK1 also promotes phosphorylation of ATG13 and FIP200 subunits and dissociate from mTOR. PI3K class III (PI3KC3), BECN1, ATG14, p150, and AMBRA1, form a protein complex (PI3KC3 complex) and initiates phagophore formation in the proximity of the ER or other membrane sources. ATG5-ATG12 complex and LC3 are involved in double-membrane vesicle (autophagosome) formation, elongation, and closure. A ubiquitin-like reaction involving ATG7 and ATG10 contributes to the conjugation of ATG12 to ATG5, which eventually interacts with ATG16 for generating a “large complex”. While ATG4 cleaves LC3 (a light chain of the microtubule-associated protein) to form LC3-I, the “large complex” acts as a ubiquitin ligase to conjugate LC3-I with the polar head of phosphatidylethanolamine (PE) to produce LC3-II, a process essential for autophagosome formation and cargo recognition. The ATG5-ATG12 complex further interacts with ATG16L1 to establish the ATG12-ATG5-ATG16L1 complex. Autophagosomes are sequentially fused with lysosomes to form autolysosomes, whose contents are subsequently digested by lysosomal hydrolases and released for metabolic recycling.

**Figure 2 cells-10-02672-f002:**
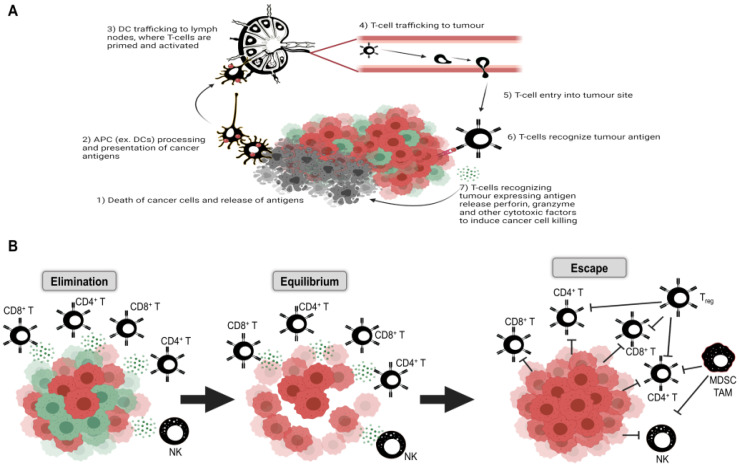
Cancer immunity cycle and immunoediting. (**A**) Cancer immunity cycle, as outlined by Chen and Mellman [161]. (**B**) Steps in the cancer immunoediting concept, with immune cell elimination of cancer cells; cancer-immune equilibrium; and escape through a multitude of immunosuppressive mechanisms (image created with BioRender.com, accessed on 21 September 2021).

**Figure 3 cells-10-02672-f003:**
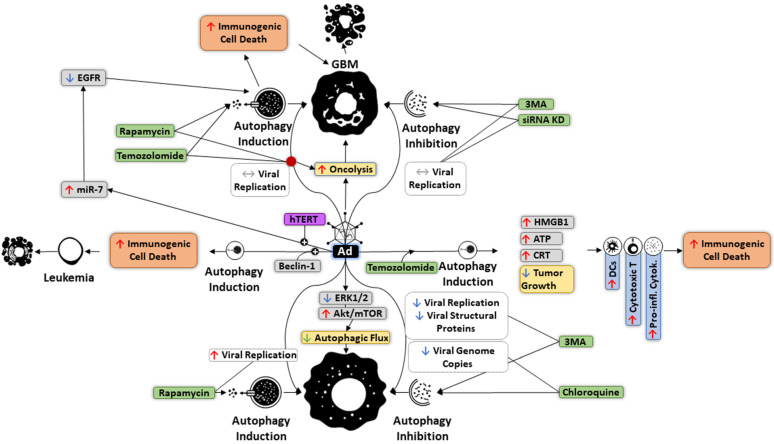
Effect of autophagy on the replication of oncolytic adenovirus (Ad). Although the entry of Ad decreases autophagic flux by affecting Erk1/2 and Akt/mTOR pathways, induction of autophagy with rapamycin can promote viral replication in cancer cells. Arming Ad with BECN1 and hTERT promoters results in autophagic cell death through different mechanisms in glioblastoma (GBM) and leukemia, respectively. Moreover, Ad-mediated oncolysis in GBM can be boosted by autophagy activation without significant impact on viral replication. Combination of Ad and temozolomide induces autophagy, which mediates immunogenic cell death via eliciting dendritic cells, cytotoxic T lymphocytes, and pro-inflammatory cytokines.

**Figure 4 cells-10-02672-f004:**
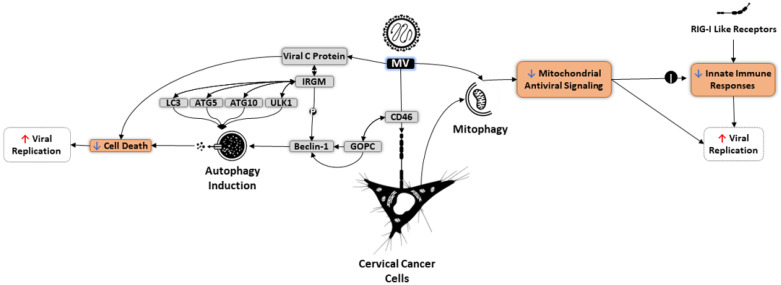
Mechanisms of autophagy induction and mitophagy modulation by oncolytic measles virus (MV). MV binds CD46 on the surface of HeLa cells. Interactions between CD46 and GOPC stimulate BECN1, contributing to autophagy induction. On the other hand, the C protein of MV interacts with IRGM, which subsequently influences several autophagy regulators such as LC3, ATG5, ATG10, and ULK1, to activate autophagy. During this process, C viral protein postpones cell death to improve MV replication and production of progeny viruses. In addition, MV uses the intact mitophagy process to inhibit mitochondrial antiviral signaling (MAVS), a mechanism that not only promotes viral replication but also limits innate immune responses against MV.

**Figure 5 cells-10-02672-f005:**
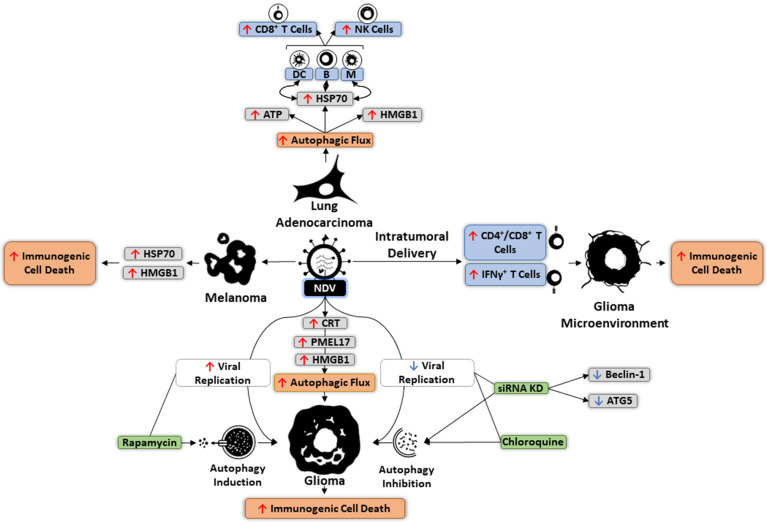
Induction of immunogenic cell death by Newcastle disease virus (NDV)-modulated autophagy. In lung adenocarcinoma cells, increased level of autophagy following NDV entry upregulates HSP70, which mediates interactions with the receptors on dendritic cells (DC), B cells, and macrophages (M). Such interactions activate natural killer cells (NK) and CD8^+^ T cell responses. In melanoma cells, NDV-induced immunogenic cell death involves upregulation of HSP70 and HMGB1. This type of cell death can also be elicited in NDV-infected glioma that occurs by overexpression of CRT, PMEL17, and HMHB1, which elevates autophagy. Rapamycin-induced promotion of autophagy in these cells increases NDV replication. In mice models, intratumoral delivery of NDV boosts the infiltration of IFNγ^+^ T cells toward the glioma microenvironment.

**Figure 6 cells-10-02672-f006:**
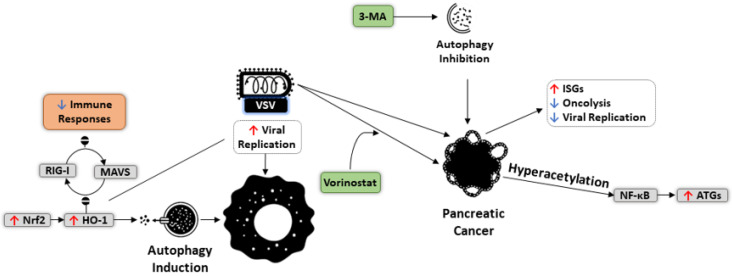
Impact of autophagy induction on the replication of oncolytic vesicular stomatitis Indiana virus (VSV). Activation of autophagy by Nrf2 pathways not only suppresses immune responses by blocking RIG-1-MAVS interactions but also elevates VSV replication in cancer cells. Combining VSV with vorinostat in pancreatic cancer cells can increase the hyperacetylation of NF-κB, ultimately promoting the expression of *ATG* genes.

**Table 1 cells-10-02672-t001:** Dual role of autophagy in tumor promotion and inhibition.

Model	Autophagy Role	Mechanism	Reference
Neuroblastoma (BE(2)-C and BE(2)-M17)	Inhibition	Degradation of gastrin-releasing peptide and inhibition of angiogenesis	[75]
Cervical Cancer (HeLa)	Promotion	Activating HIF-1α and increasing drug resistance (paclitaxel)	[76]
Gastric Cancer (MGC-803 and SGC-7901)	Inhibition	Downregulating HIF-1α and decreasing metastasis and glycolysis	[77]
Oral Cancer (SCC-9)	Inhibition	Suppressing the NF-κB pathway and inhibiting invasiveness	[78]
Bladder Cancer (T24)	Promotion	Increasing HIF-1α expression and counteracting gemcitabine-induced apoptosis	[79]
Pancreatic Cancer (PDAC)	Promotion	Degrading MHC-1 and boosting immune evasion	[80]
Breast Cancer (MCF-7)	Inhibition	Blocking nitric oxide generation and inducing apoptosis	[81]
Breast Cancer (D2A1 and MCF-7)	Promotion	Improving survival and metastasis	[82]
Hepatocellular Carcinoma (BEL7402 and HepG2)	Promotion	Elevating invasiveness via upregulating EMT markers (E-cadherin, CK18, and fibronectin)	[83]
Hepatocellular Carcinoma (SMMC-7721 and HepG2)	Promotion	Enhancing glycolysis and metastasis via upregulating MCT1 and activating Wnt/β-catenin	[84]

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
