# Peer review of "Autophagy in Tumor Immunity and Viral-Based Immunotherapeutic Approaches in Cancer"

_cells, 2021, doi:10.3390/cells10102672_

Round 1
Reviewer 1 Report
The review submitted by Zahedi-Amiri et al. is about the role of autophagy in cancer immunotherapy, with a special focus on combinatorial therapy with oncolytic viruses. The review is well written, although some parts lack references.
The following aspects require further elaboration:
- Lines 29-38, please refer to the recent literature
- Chapter 4 describes the immune editing process, a schematic and representative figure would be of help for readers
- Chapter 5 is about Autophagy and OVs, however, a proper description of the OV-cancer cell death mechanisms is missing
- The authors state that autophagy plays a dual role in cancer. A table summarizing the different roles of autophagy in various cancers, including the mechanism of action, would be beneficial for readers.
- Autophagy also leads to resistance to oncolytic virotherapy, authors should report and discuss such information
Author Response
September 21, 2021
Journal of Cells
We thank the editorials and reviewers for their positive assessment of our manuscript. Below is a point-by-point response to each of the reviewers’ comments, followed by a list of the incorporated changes we made to adhere to the editorial checklist provided.
Sincerely,
Behzad Yeganeh, PhD
Point-by-Point Response to Reviewers’ Comments
Reviewer comments:
Reviewer #1
Lines 29-38, please refer to the recent literature.
Answer: We appreciate the reviewer’s comment. We have provided references.
Chapter 4 describes the immune editing process, a schematic and representative figure would be of help for readers.
Answer: Thank you very much for the comment. We have added a representative figure in the revised manuscript.
Chapter 5 is about Autophagy and OVs, however, a proper description of the OV-cancer cell death mechanisms is missing.
Answer: We appreciate the reviewer’s comment. The main target of the manuscript is to discuss potentials for targeting autophagy as a mechanism to boost oncolytic immunotherapy. Cell death induction mechanisms by OVs are both virus-specific and cell-specific. Therefore, such broad and varied mechanisms cannot be discussed here in connection with this manuscript. However for clarification, we have provided further information which is added in the revised manuscript (Line 408-original version)
The authors state that autophagy plays a dual role in cancer. A table summarizing the different roles of autophagy in various cancers, including the mechanism of action, would be beneficial for readers.
Answer: Thank you very much for the valuable comment. We have added a representative table (Table 1) summarizing the dual role of autophagy in tumor promotion and inhibition.
Autophagy also leads to resistance to oncolytic virotherapy, authors should report and discuss such information
Answer: We appreciate the reviewer’s comment. The autophagy-induced resistance of tumor cells to oncolytic virotherapy occurs by the tumor promotion and nourishment aspect of autophagy and does not module a common mechanism across different cancer cells. Such a phenomenon depends on the stage of tumor and the type of cancer. As mentioned in the manuscript, inhibition of autophagy was found to sensitize tumor cells to different OVs (e.g., BECN1 KD, line 494-original version). Also, we have already discussed M1 OV in UPR targeting section as another example where autophagy limits the oncolytic capacity (line 565-original version). Only a few references are available with regards to Ad. For more clarification, we have provided further information which is added in the revised manuscript (Line 433-original version)
Reviewer 2 Report
The authors performed a comprehensive review covering molecular mechanism and regulation of autophagy, autophagy in tumorigenesis, autophagy in tumor immunity, autophagy and OVs, targeting autophagy for immunotherapy, and targeting autophagy in oncolytic immunotherapy. This review helps to understand recent advances in targeting autophagy in oncolytic immunotherapy.
From a clinical point of view, it is interesting to use autophagy-modifying reagents to enhance the antitumor capacity of OVs. In this review, rapamycin, rapamycin analog everolimus (Chen-G et al. Am J Physiol Cell Physiol 2019, 317, C244), vorinostat (page 12. line 491) and other autophagy inducers (Yoshida-GJ J Hematology Oncology 2017, 10, 67) as combination drug of OVs are listed. As the authors show, the role of autophagy in oncolytic immunotherapy depends on the cell type and virus used. However, in the absence of OVs, it is necessary to know in advance the effect of these drugs on the systemic immune system of cancer patients and tumor-bearing animals. It is also useful to explain how TME immune cells such as lymphocytes, DC, Treg, TAMs and NK cells are affected by these autophagy-inducing reagents.
“Section 6. Targeting autophagy for immunotherapy” can be moved before “5. Autophagy and OVs”.
A review paper entitled “Crosstalk between oncolytic viruses and autophagy” was published with the similar intent as this paper (ref. 151). Many clinical trials have recently been conducted to investigate the combined effects of OVs and immune checkpoint inhibitors, though additional effects by combination are still limited. Indeed, the authors described passive immunotherapies using monoclonal antibody in the section 4. Autophagy in tumor immunity (page 6, line 264). To clarify novel points in this paper compared to previous reviews, it is better to add a description of whether autophagy affect the expression of immune checkpoint proteins such as PD-1/PD-L1 and CTLA-4 and their blocking theraphies.
Page 9, line 420: The authors stated that “Chloroquine-mediated inhibition of autophagy in these cells has been found to reduce Ad genome copies [153]”. However, the paper [153] stated that “Treatment with two inhibitors of autophagy, chloroquine and 3-methyladenine, increased the cytotoxic effects of dl922-947 in vitro”. This discrepancy needs to be explained.
Figure 2: The arrows overlap with Cytotoxic and Pro. This should be rectified.
Author Response
September 21, 2021
Journal of Cells
We thank the editorials and reviewers for their positive assessment of our manuscript. Below is a point-by-point response to each of the reviewers’ comments, followed by a list of the incorporated changes we made to adhere to the editorial checklist provided.
Sincerely,
Behzad Yeganeh, PhD
Point-by-Point Response to Reviewers’ Comments
Reviewer comments:
Reviewer #2
From a clinical point of view, it is interesting to use autophagy-modifying reagents to enhance the antitumor capacity of OVs. In this review, rapamycin, rapamycin analog everolimus (Chen-G et al. Am J Physiol Cell Physiol 2019, 317, C244), vorinostat (page 12. line 491) and other autophagy inducers (Yoshida-GJ J Hematology Oncology 2017, 10, 67) as combination drug of OVs are listed. As the authors show, the role of autophagy in oncolytic immunotherapy depends on the cell type and virus used. However, in the absence of OVs, it is necessary to know in advance the effect of these drugs on the systemic immune system of cancer patients and tumor-bearing animals. It is also useful to explain how TME immune cells such as lymphocytes, DC, Treg, TAMs and NK cells are affected by these autophagy-inducing reagents.
Answer: Thank you very much for the valuable comment. Further information on Effect of autophagy-modulating agents on immune cells has been added in the revised manuscript (after Line 557-original version).
“Section 6. Targeting autophagy for immunotherapy” can be moved before “5. Autophagy and OVs”.
Answer: We appreciate the reviewer’s comment. This has been fixed in the revised manuscript
A review paper entitled “Crosstalk between oncolytic viruses and autophagy” was published with the similar intent as this paper (ref. 151). Many clinical trials have recently been conducted to investigate the combined effects of OVs and immune checkpoint inhibitors, though additional effects by combination are still limited. Indeed, the authors described passive immunotherapies using monoclonal antibody in the section 4. Autophagy in tumor immunity (page 6, line 264). To clarify novel points in this paper compared to previous reviews, it is better to add a description of whether autophagy affect the expression of immune checkpoint proteins such as PD-1/PD-L1 and CTLA-4 and their blocking theraphies.
Answer: Thank you very much for the valuable comment. Further information on Effect of autophagy on expression of immune checkpoints has been added in the revised manuscript (added to Line 555-original version).
Page 9, line 420: The authors stated that “Chloroquine-mediated inhibition of autophagy in these cells has been found to reduce Ad genome copies [153]”. However, the paper [153] stated that “Treatment with two inhibitors of autophagy, chloroquine and 3-methyladenine, increased the cytotoxic effects of dl922-947 in vitro”. This discrepancy needs to be explained.
Answer: Several studies suggest that increased level of cytotoxicity does not necessarily correspond to successful/complete stages of viral replication. Based on some observations, only binding and entry of some viruses were found to be efficient enough to activate various cell death mechanisms. The cited paper itself refers to an older study where autophagy inhibition by 3-MA restricts the production of adenoviral structural proteins (https://www.sciencedirect.com/science/article/pii/S0042682211001978). Also, their data on the cleaved caspase 3 allowed them to discuss the activation of apoptosis along with the suppression of autophagy, which makes cancer cells more vulnerable to OV, despite limited viral genome copies. They supported this hypothesis in their animal studies as well because apoptosis was not seen to be activated in either chloroquine-treated or Ad-treated tumors, suggesting a cytotoxic impact that is only induced when Ad infection is associated with autophagy inhibition.
Figure 2: The arrows overlap with Cytotoxic and Pro. This should be rectified.
Answer: We appreciate the editorial’s comment. We have fixed this in the revised manuscript

Round 2
Reviewer 1 Report
The authors provided satisfactory replies and improvements.